# Effects of Grain Refinement and Thermal Aging on Atomic Scale Local Structures of Ultra-Fine Explosives by X-ray Total Scattering

**DOI:** 10.3390/ma15196835

**Published:** 2022-10-01

**Authors:** Jiangtao Xing, Weili Wang, Shiliang Huang, Maohua Du, Bing Huang, Yousong Liu, Shanshan He, Tianle Yao, Shichun Li, Yu Liu

**Affiliations:** 1College of Ordnance Engineering, Naval University of Engineering, Wuhan 430033, China; 2China Academy of Engineering Physics, Institute of Chemical Materials, Mianyang 621900, China; 3Navy Research Institute, Beijing 100072, China

**Keywords:** ultra-fine explosive, defect, selectivity of the sensitivity, total scattering, pair distribution function

## Abstract

The atomic scale local structures affect the initiation performance of ultra-fine explosives according to the stimulation results of hot spot formation. However, the experimental characterization of local structures in ultra-fine explosives has been rarely reported, due to the difficulty in application of characterization methods having both high resolution in and small damage to unstable organic explosive materials. In this work, X-ray total scattering was explored to investigate the atomic scale local distortion of two widely applicable ultra-fine explosives, LLM-105 and HNS. The experimental spectra of atomic pair distribution function (PDF) derived from scattering results were fitted by assuming rigid ring structures in molecules. The effects of grain refinement and thermal aging on the atomic scale local structure were investigated, and the changes in both the length of covalent bonds have been identified. Results indicate that by decreasing the particle size of LLM-105 and HNS from hundreds of microns to hundreds of nanometers, the crystal structures remain, whereas the molecular configuration slightly changes and the degree of structural disorder increases. For example, the average length of covalent bonds in LLM-105 reduces from 1.25 Å to 1.15 Å, whereas that in HNS increases from 1.25 Å to 1.30 Å, which is possibly related to the incomplete crystallization process and internal stress. After thermal aging of ultra-fine LLM-105 and HNS, the degree of structural disorder decreases, and the distortion in molecules formed in the synthesis process gradually healed. The average length of covalent bonds in LLM-105 increases from 1.15 Å to 1.27 Å, whereas that in HNS reduces from 1.30 Å to 1.20 Å. The possible reason is that the atomic vibration in the molecule intensifies during the heat aging treatment, and the internal stress was released through changes in molecular configuration, and thus the atomic scale distortion gradually heals. The characterization method and findings in local structures obtained in this work may pave the path to deeply understand the relationship between the defects and performance of ultra-fine explosives.

## 1. Introduction

Ultra-fine explosives with a particle size around hundreds of nanometers show both high safety and high reliability, and have been widely applied in aerospace, military, and industry. For example, ultra-fine 2,6-diamino-3,5-dinitropyrazine-1-oxide (LLM-105) [1] and ultra-fine 2,2′,4,4′,6,6′-hexanitro diphenylethylene (HNS) [2] show low sensitivity to impact, but high-sensitivity, short-duration pulsed shock waves with high pressure, which is called selectivity of the sensitivity [3], and have been used as advanced initiating explosives in detonators. The selectivity of the sensitivity of ultra-fine explosives is the key to their performance in applications. According to the hot spot theory, the selectivity of the sensitivity is highly related to the size and number of defects in explosives [4,5]. A lot of numerical simulation works have reported the process of forming hot spots from defects in explosives such as atomic scale local distortion structures and nano pores [6,7,8,9,10,11,12,13,14,15,16]. Armstrong and Coffey [17,18] revealed that the dislocation region releases a large amount of heat under shock, and promotes the formation of hot spots, indicating the important role of local distortion structures for sensitivity. However, the distortion structures used in simulations were assumptive, which results in deviation between simulation and reality. Therefore, it is necessary to carry out the experimental characterization of the atomic scale local distortion of ultra-fine explosives, which has been rarely reported.

The commonly used characterization method for atomic scale distortion in materials is high resolution transmission electron microscope (HRTEM), which plays an important role in the research of inorganic materials. However, TEM cannot be directly used to characterize explosives because of the damage from electron beams [19,20,21]. In recent years, total scattering measurements and atomic pair distribution function (PDF) analysis [22,23,24] have attracted attention in the field of condensed matter because of the advantages in high resolution for studying atomic scale structures and small damage to samples. PDF analysis is a powerful approach to study short- and intermediate-range order in materials on the nanoscale. It may be obtained from total scattering measurements using X-rays, neutrons, or electrons, and it provides structural details when defects, disorder, or structural ambiguities obscure their elucidation directly in reciprocal space. PDF analysis was originally developed to study the structure of amorphous, liquid, and other short-range ordered materials. At present, it has been widely used for nanocrystalline structures and atomic scale local structures of inorganic materials [25,26,27,28,29,30]. However, there are few examples of applying PDF analysis in organic materials because of the complex structures of organic molecules and crystals leading to the difficulty in result interpretation [31,32,33,34].

In this work, X-ray total scattering was explored to investigate the atomic scale local distortions of ultra-fine LLM-105 and ultra-fine HNS. The assumption of rigid ring structure in LLM-105 and HNS molecule was used for fitting the experimental PDF spectra that gives slight changes in molecular configuration. The effects of grain refinement and thermal aging on the atomic scale local structure were investigated, and the changes in both the length and angle of covalent bonds have been identified. The characterization method and findings in local structures obtained in this work may pave the path to deeply understand the relationship between the defects and performance of ultra-fine explosives.

## 2. Materials and Methods

### 2.1. Materials

Raw LLM-105 and HNS were provided by the Institute of Chemical Materials. Ultra-fine LLM-105 and HNS were prepared by the antisolvent recrystallization method. Dimethyl sulfoxide and N,N-Dimethylformamide were selected as the solvents for preparing ultra-fine LLM-105 and NHS, respectively, and the anti-solvent of both was deionized water. The optical microscope after simple grinding of raw LLM-105 and HNS is shown in Figure 1a,b, and the SEM images of ultra-fine LLM-105 and HNS is shown in Figure 1c,d. The particle sizes of raw LLM-105 and raw HNS after simple grinding are about 60 µm and 200 µm, respectively. Ultra-fine LLM-105 and ultra-fine HNS with particle sizes about 500 nm and 300 nm, respectively. 

### 2.2. Experimental Process

Under normal temperature and pressure, the explosive powder sample was uniformly filled into a quartz glass tube with a thickness of 1 mm, and then placed on the test bench. The particle size of raw materials is large, and it is difficult to fill the raw materials in a 1 mm thick glass tube. Therefore, in the experiment, the raw materials were simply ground to facilitate loading. Shanghai synchrotron radiation facility (SSRF) is composed of a 3.5 GeV electron storage ring, 3.5 GeV booster, and a 150 MeV linac. The beam current in the storage ring is 240 mA. The BL14B1 beamline has three critical components: a collimating mirror, a sagittally focused double crystal monochromator, and a focusing mirror, which are 21, 24, and 30 m away from the source, respectively. The X-ray energy of BL14B1 beamline is 18 keV, and the wavelength is λ = 0.6889 Å [35]. Mythen 1K linear detector [36] was selected as the detector. The 2θ scanning range was 1~81.53° with step of 0.00398°. It is worth noting that the total scattering signal obtained after the sample test is normalized, and then the total scattering signal of the empty capillary is deducted. 

A diffractometer [36] was used for the PDF experiment and the Q resolution is generally very high, which should be lower than 0.1% given the relatively low energy and extremely high angle resolution in the Mythen detector (approximately 0.004°). The Qdamp was set to 0.015 and Qbroad was set to be zero in a fitting *r* range lower than 10 Å. Usually for high crystalline material, the too low cutoff Qmax leads to a larger broadening of peaks and this make the Qbroad parameter unprecise in calibration, although calibration of Qbroad for low Qmax has been realized [37,38].

When software GudrunX [39] was used to process the experimental data, the relevant parameters (‘Incident beam polarization factor’ and ‘Factor for compton scattering’) were optimized to obtain the structure factor and PDF curve of sample. The PDFgui 1.0 software was used to fit the experimental PDF curve [40].

For thermal aging, the heating temperature was 120 °C, and the aging time was 30 days. Experimental studies have found that the particles of ultra-fine explosives increase after heating at 120 °C for 30 days, and the specific surface area decreases significantly [41]. In order to explore its growth mechanism, relevant thermal aging experiments were carried out.

## 3. Results and Discussion

### 3.1. The Effects of Grain Refinement

The PDFgui software was used to calculate the PDF of LLM-105 and HNS by using the CIF files [42,43], and the results are shown in Figure 2.

As seen from Figure 2a,b, in the range of 1 to 10 Å, 16 peaks are shown for LLM-105, indicating that the atom pair interaction information is rich. In the same range, 20 peaks are shown for HNS, and the distribution of peaks is compact in the range of 4 to 10 Å. The reason for the difference is that HNS has a larger molecular weight and closer distribution distance between different atoms in the molecule compared with LLM-105. The simulation results show that the total scattering PDF can achieve the explosive structure characterization resolution of Å. For materials with long-range order, the atomic scale local structures can be obtained by analyzing the overall average structure characteristics.

LLM-105 and HNS with different particle sizes were characterized by a total scattering test. The structure factor of the sample is obtained by processing the data, as shown in Figure 3. It can be seen that the structural factor strength of ultra-fine LLM-105 and HNS is lower than that of raw materials, which is caused by the decrease of crystallinity after the grain refinement of explosives. In Figure 3a,b, there are several peaks with overestimated strength, which may be caused by the orientation of raw material samples in glass tubes.

The PDF transforms the structure information of material into the radial pair distribution of different bond length scales in real space, so that the unit cell information in traditional crystallography is expressed by the bond length of atomic pair. Therefore, the short-range and local structure information would not be masked by the long-range structure. In the atomic pair distribution function, the peak position represents the atomic pair spacing or bond length, the peak area represents the coordination number deducting the average number density, and the peak width represents the degree of structural disorder (including static disorder, thermal disorder, and intrinsic broadening of the test). The atomic pair peak attenuation in the radial range is closely related to the spatial scale of the ordered structure of condensed matter, and the nearest neighbor coordination structure in the short-range scale corresponds to the atomic pair peak one by one.

To gain insight into how intramolecular and intermolecular atom–atom distances appear in the PDF, we define the largest intramolecular atom–atom distance as *l* and the shortest intermolecular atom–atom distance as *k*. For the case of LLM-105, *k* ≈ 2.7 Å. In the *r* region below *k*, the PDF curve contains only intramolecular atom–atom distances with sharp narrow peaks. In the high-*r* region beyond *l* ≈ 6.7 Å, the PDF curve contains only intermolecular atom–atom distances, which yield broader peaks in the PDF curve. In the region between *k* and *l*, the sharp intramolecular peaks and broad intermolecular peaks coexist. For HNS, there is a similar law. The shortest intermolecular atom–atom distance is *k* ≈ 3.0 Å, and the maximum intramolecular atom–atom distance is *l* ≈ 13.6 Å.

The PDF curves obtained for explosives with different particle sizes are shown in Figure 4. The peak within the range of 1~2 Å is the strongest and the sharpest, which can be seen from Figure 4a,c. This is because the correlated motion of atoms: strongly bonded atoms tend to move together, dependent on each other. These motional correlations tend to die out smoothly with increasing distance [44]. It can be seen from Figure 4a that the PDF curve of ultra-fine LLM-105 is similar to that of raw LLM-105. Figure 4c shows that, the difference of PDF curve between raw and ultra-fine HNS is small in the range of 1–4.5 Å, and obvious in the range of 4.5–10 Å. Compared with LLM-105, the PDF curve changes more obviously after HNS grain refinement. This is because the particle size of HNS refined is smaller and the HNS molecule is larger. The refining method used in this study is recrystallization, in which the molecular arrangement is from disorder to order. The ultra-fine HNS particle size is smaller than LLM-105, which means that the degree of crystallization is lower and the degree of disorder is larger. At the same time, the HNS molecule is larger than LLM-105, which means that the energy barrier of molecular rotation and configuration adjustment is higher. For the above two reasons, the change of PDF curve after HNS grain refinement is more obvious than that of LLM-105. The peak positions of ultra-fine LLM-105 are shifted to the left compared with raw LLM-105 within 1–2.7 Å. The peak area is reduced, and the peak width is widened. For example, the peak at 1.25 Å position moves from 1.25 Å to 1.15 Å (the peak at 1.25 Å corresponds to the average length of covalent bonds within the molecule), the peak width increases from 0.36 to 0.41, and the peak area decreases from 1.38 to 1.27. The results show that the distance between adjacent atomic pairs in the molecule decrease after LLM-105 is refined, and the degree of disorder increases. The decrease of atomic pair distance indicates that the interaction between atoms is enhanced. From Figure 4c, it can be found that the peak positions of ultra-fine HNS within 1–3 Å are shifted to the right compared with the raw HNS. For example, the peak at 1.25 Å moves from 1.25 Å to 1.30 Å, and the peak area is reduced from 1.11 to 0.97. It shows that the reduction of particle size can increase the distance between adjacent atomic pairs in HNS molecule and weaken the interaction between atoms. The atomic scale distortion of LLM-105 and HNS shows different trends after grain refinement, which possibly reflects different structural state of the two explosives in solutions of recrystallization process.

The PDF curve obtained from the experiment was fitted by PDFgui based on the crystal structure parameters of the two explosives. This program is used to fit the structure by moving a single atom, which is very useful for inorganic compounds. However, when fitting organic compounds with defined molecular composition, constraints need to be introduced to maintain the length of covalent bonds and rigid atom group that is usually called rigid body constraints in crystallography programs [33]. For LLM-105 molecule, the pyrazine ring is rigid, and the stacking state of molecules inside the lattice does not change. The local structure distortion of LLM-105 possibly occurs on nitro and amino groups. For HNS, the benzene ring structure is rigid, but there may be relative displacement and deflection between the two benzene rings. Therefore, during the fitting process, the structures of both pyrazine rings and benzene rings were fixed. The fitting results in Figure 4b,d show that *R_w_* of LLM-105 is much lower than HNS, indicating better fitting of LLM-105, which is possibly caused by the smaller molecular of LLM-105 than HNS. The PDF fitting of complex organic molecules is still a challenge [32].

The parameters of LLM-105 crystal structure model are shown in Table 1. The lattice parameters of ultra-fine LLM-105 do not change significantly compared with the raw materials.

For inorganic materials, the type of atomic pairs corresponding to each peak position in the PDF curve can be determined according to the distance between each atomic pair in the lattice of inorganic materials. The lattice structure of inorganic materials is relatively simple, and there are few kinds of atoms, so it is easy to find the types of atomic pairs corresponding to each peak. In this work, both explosives are composed of four types of atoms (C,H,O,N). Due to the low resolution of X-ray to H atom, there are 6 types of atom pairs except H atom, which are C—C, O—O, N—N, C—O, C—N, O—N. Moreover, the simplified atomic pair distribution function G(r) is formed by the superposition of various types of atomic PDF. Therefore, the types of atomic pairs corresponding to each peak position in Figure 4a,c are difficult to be determined. 

In order to investigate the changes of the distance between atom pairs of LLM-105 after grain refinement, the PDF curves of adjacent N—N atoms on both sides of LLM-105 molecule and O—O atom pairs in nitro group were calculated separately according to the structural model parameters and anisotropic atomic displacement parameters (ADPs) obtained by fitting, as shown in Figure 5a,b. The corresponding atomic parameters were labeled in Figure 5c. The red circle in the figure indicates that the structure of pyrazine ring is fixed in LLM-105.

It can be found in Figure 5a,b that the peak width of the PDF curves of LLM-105 after grain refinement is generally widened. Taking the peak at 2.95 Å of N_2_—N_3_ in Figure 5a as an example, the peak width increases from 0.285 to 0.354, which shows that the disorder degree increases after LLM-105 is refined. The distance of N_2_—N_3_ atoms in the molecule increases from 2.95 Å to 3.03 Å, and the distance of N_5_—N_6_ in the molecule increases from 2.97 Å to 3.07 Å. Although the distance of O_2_—O_3_ atom pairs decreases from 2.20 Å to 2.12 Å, the distance of O_4_—O_5_ atoms decreases from 2.20 Å to 2.05 Å. The results show that the distance of N_2_—N_3_ and N_5_—N_6_ in LLM-105 tends to increase and the interaction force decreases, whereas the distance of O_2_—O_3_ and O_4_—O_5_ atomic pairs decreases and the atomic interaction force increases with the decrease of particle size. The changes in PDF curves of N—N and O—O pairs indicate the variations in angles of covalent bonds. The structures of the molecules were distorted possibly due to the incomplete crystallization process and internal stress.

To sum up, after LLM-105 is refined, the average length of covalent bonds reduces by 0.1 Å, the interaction between atomic pairs enhanced, and the disorder degree increased. It is worth noting that the molecular configuration of LLM-105 also changed after grain refinement. The distance of adjacent N—N atoms on both sides of LLM-105 molecule increases, whereas the distance of O—O atoms in nitro group decreases, which describes the variations in angles of covalent bonds. After HNS is refined, the average length of covalent bonds increases by 0.05 Å, and the disorder degree increases. Therefore, the total scattering technique can be used to characterize the atomic scale local distortion of ultra-fine explosives. 

### 3.2. The Effects of Thermal Aging

The ultra-fine LLM-105 and HNS aged at 120 °C for 30 days were characterized by total scattering. The data were processed to obtain the structure factor and PDF curve of the sample and compared with the ultra-fine LLM-105 and HNS samples before aging. The structure factor of the sample is shown in Figure 6, and the PDF curve is shown in Figure 7a,c.

The variation range of each peak in the PDF curve is small, and the overall trend of PDF curve is similar after heat aging treatment of ultra-fine LLM-105 and HNS. The peak positions of ultra-fine LLM-105 after heating aging shift to the right within 1–2.7 Å. For example, the peak position at 1.15 Å moves to 1.27 Å and the peak width decreases from 0.46 to 0.44, which indicates that the average covalent bond distance in the molecule increases, the interaction force becomes weak, and the disorder degree decreases. For HNS, the results are opposite to LLM-105. Compared with the HNS before aging, the peak areas of ultra-fine HNS after thermal aging within 1–3 Å generally increased. For example, for the peak of covalent bonds, the peak position moves from 1.30 Å to 1.20 Å, the peak area increases from 1.19 to 1.56, and the peak width decreases from 0.37 to 0.35. This shows that after ultra-fine HNS thermal aging treatment, the distance between adjacent atomic pairs in the molecule decreases, the interaction between atoms increases, the number of defects decreases, disorder degree decreases, and atomic scale defects gradually heal. 

The PDF curves of the aged samples were fitted and shown in Figure 7b,d. The fitting result of LLM-105 is better than that of HNS. According to the structural model parameters obtained during the PDF curve fitting of LLM-105, the PDF curves of N—N and O—O atom pairs are calculated separately as shown in Figure 8a,b. The corresponding atomic were labeled in Figure 8c. The crystal structure parameters of LLM-105 obtained in the fitting process are shown in Table 2. It can be seen that the lattice of ultra-fine LLM-105 expands mainly along the b direction after aging.

By analyzing the atomic pairs corresponding to specific peaks in Figure 8a,b, it can be found that after thermal aging, the distance of N_2_—N_3_ atoms in ultra-fine LLM-105 molecule reduced from 3.03 Å to 2.92 Å, and the distance between N_5_—N_6_ atoms reduced from 3.07 Å to 2.97 Å. Although the distance of O_2_—O_3_ atom pairs increased from 2.12 Å to 2.23 Å, the distance of O_4_—O_5_ atoms increased from 2.05 Å to 2.15 Å. The results show that the distance of N_2_—N_3_ and N_5_—N_6_ atoms pairs in ultra-fine LLM-105 decreases and the interaction force between atoms increases after thermal aging treatment. However, the distance of O_2_—O_3_ and O_4_—O_5_ atomic pairs increases and the interaction between atoms decreases. These results show that the change trend of covalent bond angle caused by aging is opposite to the result of refining, indicating that the distortion formed during refining heal during aging.

To sum up, after thermal aging, the average covalent bond distance in ultra-fine LLM-105 molecule increases by 0.12 Å, and the disorder degree decreases. It is worth noting that the molecular configuration also changed after thermal aging of LLM-105. The distance of adjacent N—N atoms on both sides of LLM-105 molecule decreased, whereas the distance between O—O atoms in nitro group increased. For HNS, heating can reduce the covalent bond length by 0.1 Å, and the disorder degree decreases. Atomic scale defects gradually heal, thus reducing the number of defects. This result is contrary to the change rule of local structure obtained in the previous section. This is because, after the grain refinement of LLM-105 and HNS, the structures of the molecules were distorted possibly due to the incomplete crystallization process and internal stress. During the heat aging treatment, the atomic vibration in the molecule intensifies, and the internal stress was released through changes in molecular configuration, and thus the atomic scale distortion gradually heals.

Although it is difficult to relate the atomic scale local structure of ultra-fine explosives obtained in this paper to the application of ultra-fine explosives, this result provides support for understanding the relationship between structure and explosive performance. In the latest research, it is found that the local structure has an impact on the performance [45], but these results are based on certain simulation assumptions. Our conclusion supports the hypothesis and makes the simulation better correlated with the actual performance.

## 4. Conclusions

(1)The molecular configuration and structural disorder of ultra-fine LLM-105 and HNS have been characterized by X-ray total scattering.(2)By decreasing the particle size of LLM-105 and HNS from hundreds of microns to hundreds of nanometers through recrystallization, the degree of structural disorder increases, and both length and angle of covalent bonds of molecules changes. The average length of covalent bonds in LLM-105 reduces from 1.25 Å to 1.15 Å, whereas that in HNS increases from 1.25 Å to 1.30 Å, which is possibly related to the incomplete crystallization process and internal stress.(3)After thermal aging of ultra-fine LLM-105 and HNS, the degree of structural disorder decreases, and the distortion in molecules formed in the synthesis process gradually heals. The average length of covalent bonds in LLM-105 increases from 1.15 Å to 1.27 Å, whereas that in HNS reduces from 1.30 Å to 1.20 Å. The possible reason is that the atomic vibration in the molecule intensifies during the heat aging treatment, and the internal stress was released through changes in molecular configuration, and thus the atomic scale distortion gradually heals.

## Figures and Tables

**Figure 1 materials-15-06835-f001:**
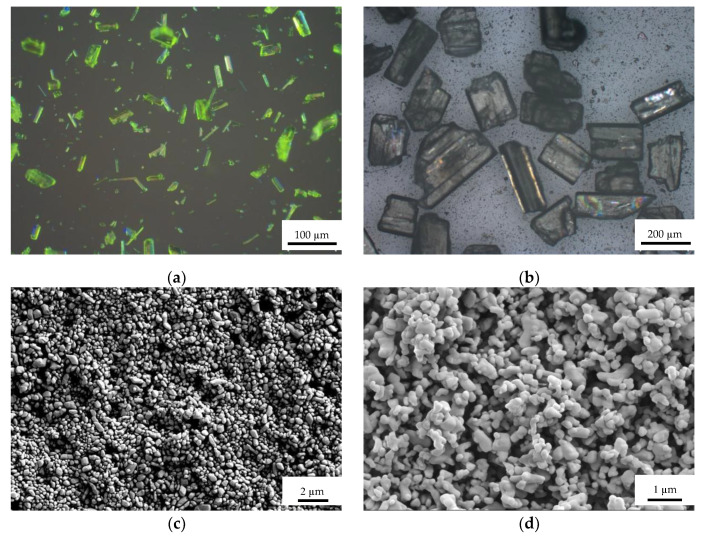
Optical microscope and SEM images of samples: (**a**) Raw LLM-105; (**b**) Raw HNS; (**c**) Ultra-fine LLM-105; (**d**) Ultra-fine HNS.

**Figure 2 materials-15-06835-f002:**
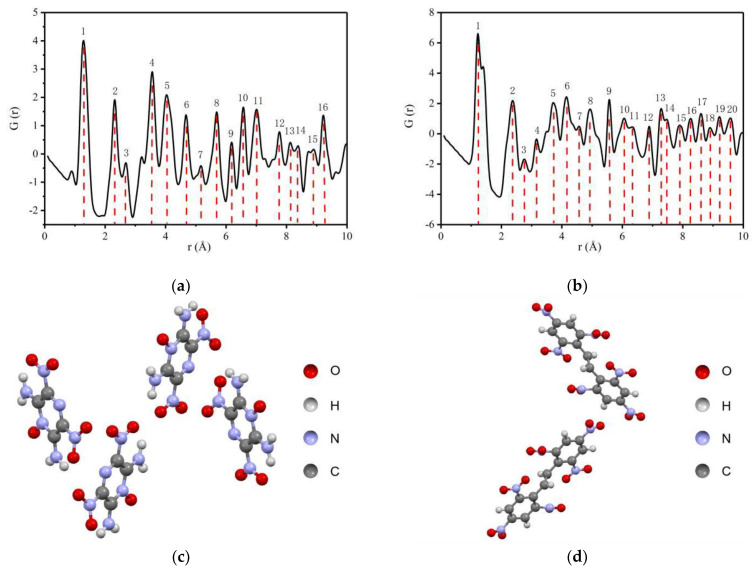
PDF curves of LLM-105 and HNS calculated by crystal structures: (**a**) PDF of LLM-105; (**b**) PDF of HNS; (**c**) Crystal structure of LLM-105; (**d**) Crystal structure of HNS.

**Figure 3 materials-15-06835-f003:**
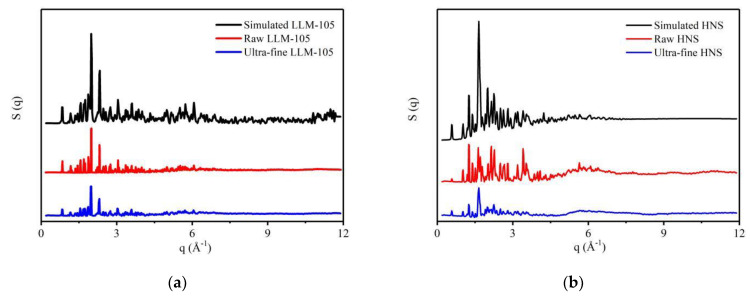
Structure factors of explosives with different particle sizes: (**a**) LLM-105; (**b**) HNS.

**Figure 4 materials-15-06835-f004:**
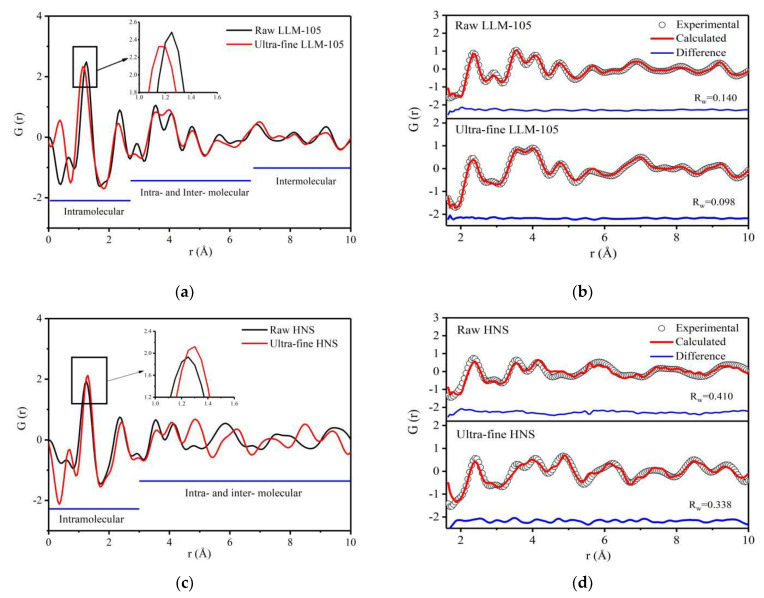
PDF curves and fitting results of two samples with different particle sizes: (**a**) PDF curve of LLM-105 with different particle sizes; (**b**) PDF fitting results of LLM-105; (**c**) PDF curve of HNS with different particle sizes; (**d**) PDF fitting results of HNS.

**Figure 5 materials-15-06835-f005:**
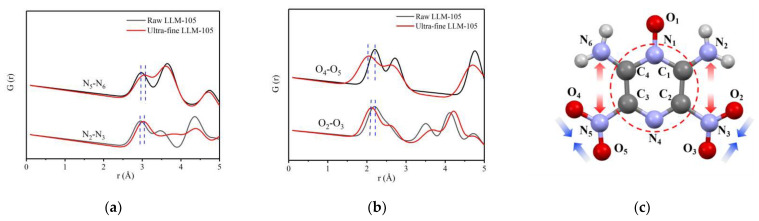
(**a**) PDF curve of N—N atom pair in LLM-105; (**b**) PDF curve of O—O atom pair in LLM-105; (**c**) Schematic diagram of changes of atomic pairs in LLM-105 molecule.

**Figure 6 materials-15-06835-f006:**
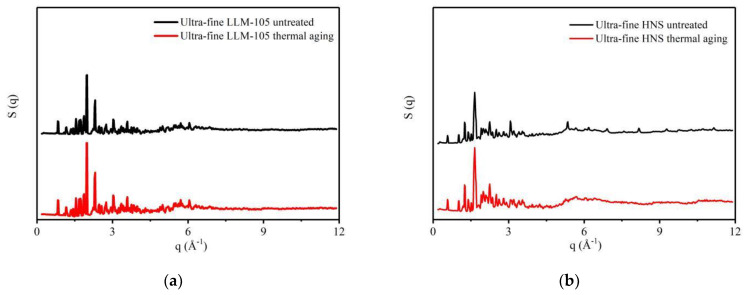
Structure factors of explosives before and after thermal aging: (**a**) LLM-105; (**b**) HNS.

**Figure 7 materials-15-06835-f007:**
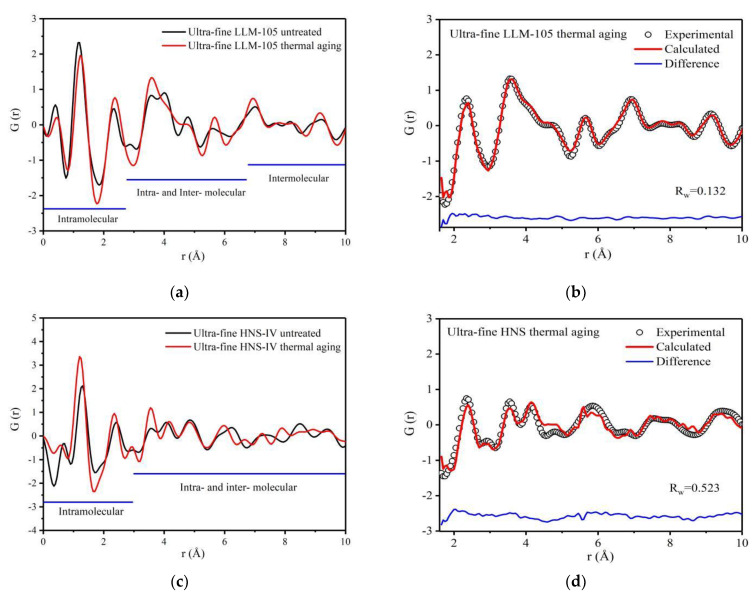
PDF curves and fitting results of two samples before and after thermal aging: (**a**) PDF curve of LLM-105 before and after thermal aging; (**b**) PDF fitting results of LLM-105; (**c**) PDF curve of HNS before and after thermal aging; (**d**) PDF fitting results of HNS.

**Figure 8 materials-15-06835-f008:**
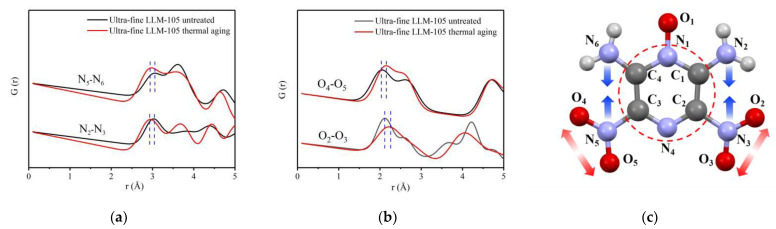
(**a**) PDF curve of N—N atom pair in LLM-105; (**b**) PDF curve of O—O atom pair in LLM-105; (**c**) Schematic diagram of changes of atomic pairs in LLM-105 molecule after thermal aging.

**Table 1 materials-15-06835-t001:** The parameters of LLM-105 crystal structure model with different particle sizes.

Sample	a (Å) *	b (Å) *	c (Å) *	α *	β *	γ *
Raw LLM-105	5.74146	15.9116	8.39469	90°	101.14°	90°
Ultra-fine LLM-105	5.75434	15.8775	8.43920	90°	101.13°	90°

* a, b, c in the cell are the lengths of unit translation vectors in the three crystal axis directions respectively. α, β, γ are the included angles between the three axes (a-b, a-c, b-c) respectively.

**Table 2 materials-15-06835-t002:** Changes of LLM-105 crystal structure parameters before and after thermal aging.

Sample	a (Å)	b (Å)	c (Å)	α	β	γ
Untreated	5.75434	15.8775	8.4392	90°	101.13°	90°
After thermal aging	5.72687	15.9715	8.4262	90°	100.97°	90°

## Data Availability

The data that support the findings of this study are available from the corresponding author upon reasonable request.

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
