# Peer review of "Effects of Grain Refinement and Thermal Aging on Atomic Scale Local Structures of Ultra-Fine Explosives by X-ray Total Scattering"

_materials, 2022, doi:10.3390/ma15196835_

Round 1
Reviewer 1 Report
Presented paper reports on application of Total X-ray scattering and Pair Distribution Function analysis for study of tiny structural transformations in two explosive materials. In general, I found it interesting, nevertheless several serious issues should be addressed.
1) line 67 ‘In this work, the X-ray total scattering with angstrom resolution was explored’ and
lines 300-302 ‘The molecular configuration and and structural disorder … have been characterized by X-ray total scattering, and the resolution reaches angstrom. ’ It is not clear from the text, what do authors mean – best resolution of collected total scattering data (i.e. d_min = lambda/( 2 sin Theta_max)) or peak-to-peak resolution of generated PDF. In case the ‘angstom resolution’ refers to resolution of peaks on PDF it seems to be incorrect to discuss the variation of interatomic distances by 0.08 – 0.15â„«. In general, the resolution of PDF used to be characterized by Qbroad parameter which depends on instrumental conditions. Authors are kindly requested to provide the PDF of reference material (Ni, LaB6, CeO2 etc.) collected using exactly the same experimental conditions and its fit to define Qbroad and Qdamp parameters.
2) Since PDF is the only experimental data analyzed in present manuscript authors are kindly requested to provide more details of total scattering experiments and PDF generation. Particularly, it is not clear if the total scattering was performed for empty capillary as a background signal and how the correction was applied. Authors should indicate if they used identical parameters for GudrunX to obtain PDF of all samples or optimized these parameters for each sample separately. There is also a discrepancy in the given experimental conditions, namely the maximum scanning angle 2theta has a value of 83.53deg, but this is impossible for an initial angle of 1deg and a step size of 0.02 deg. Please clarify.
3) Theory of PDF has been described and explained in numerous papers, including recent reviews cited in manuscript. So, I strongly suggest to exclude the section 2 ‘Theoretical of PDF’ from the maintext.
4) Lines 105-108. Authors report particle size of raw and recrystallized materials, but these values are not supported by any experimental data! Please, confirm this statement at least by optical microscopy and scanning electron microscopy or atomic force microscopy. Details of re-crystallization process are also omitted, please indicate at least a solvent.
5) Particle size of 0.4-0.5 mm seems to be too large for XRD analysis, since in 1 mm thick capillary it gives only two (!) particles in the beam, while according to good practice it should be ~ 10 000 particles within the scattering volume of sample. Insufficient number of particles and their large size resulted in incorrect determination of intensities of peaks on XRD. Indeed, several peaks with overestimated intensities are clearly visible on XRD patterns of both raw explosives (Fig. 2a,b). This certainly affects the generated PDF and should be avoided.
6) Fig 2 shows XRD patterns of raw and re-crystallized material, but XRD patterns of annealed samples are not presented. Please, add also calculated XRD patterns of LL and HNS to confirm the identity of powders and reported crystal structures.
7) Fig. 4 shows simulated PDF for four samples, while no reference on ‘theoretical profile’ are not presented.
Reviewer 2 Report
The manuscript “Characterization on Atomic Scale Local Structures of Ultra-fine Explosives by X-ray Total Scattering” by Jiangtao Xing, Weili Wang, Shiliang Huang, Maohua Du, Bing Huang, Yousong Liu, Shanshan He, Tianle Yao, Shichun Li, Yu Liu reports new results with regard of the experimental characterization of local structures in ultra-fine explosives, the issue not frequently addressed in the literature.
Some issues have to addressed all over the manuscript text.
Line 41 There is lack of space in “tohigh-pressure shock”. The term “sensitivity selectivity” should be checked if it is generally accepted designation. Also the literature reference at the end of this sentence is needed.
Line 45 I recommend substituting “simulation works” by “theoretical works” or “numerical simulation works”.
Line 58 In “…atomic pair distribution function (PDF) technology”, the PDF is a technology or method?
Line 60 What does exactly mean “Total scattering”? I think it can be explained better in the text.
Line 87 “The PDF transforms the structure information of matter into…” I suggest replacing matter by material or substance.
Line 109 The section 3.2. Experimental process is too concise in my opinion. The authors have to give more details. The word “with” appears in different font from other text. Also, pointing and writing style should be checked.
Line 115 “PDFgui” it seems there is an error or this abbreviation has to be explained better.
Lines 117-118 “For thermal aging, the heating temperature was 120 ℃, and the aging time was 30 days”. Why these temperature and time were chosen?
According to the results obtained, it should be explained clearer which gains the experimental techniques proposed and subsequent analyses brought to better use of the ulta-fine explosives LLM-105 and HNS. If is it possible to conclude from the present study which one, LLM-105 or HNS explosive, is more advantageous and for which application?
Reviewer 3 Report
The paper written by the following Authors: Jiangtao Xing, Weili Wang, Shiliang Huang, Maohua Du, Bing Huang, Yousong Liu, Shanshan He, Tianle Yao, Shichun Li, Yu Liu, entitled “Characterization on Atomic Scale Local Structures of Ultra-fine 2 Explosives by X-ray Total Scattering” presents an interesting study on atomic scale local distortion of ultra-fine LLM-105 and ultra-fine HNS.
Although the paper is interesting, I have some major concerns:
Title
The title reflects the results presented here.
Abstract
The abstract is lacking the informative conclusion. It should be written in more details.
Material and Methods
1. There is no information about the material preparation as well as applied infrastructure.
2. There is no information abut the statistical methods that should be included for the results analysis.
Reviewer 4 Report
In this paper method to analyze atomic scale local structures of ultra-fine explosives based on X-ray total scattering are presented and implemented for LLM-105 and HNS materials.
I believe this work is an interesting research that is useful for the community. However, a few remarks can be done according to the text of the manuscript.
- Line 112. Please, correct energy units spelling (keV instead of KeV).
- Line 112. Please, add some information about the source of synchrotron radiation used for experimental measurements: the name of the synchrotron and particular beamline/ endstations used; short description or link/reference to such description.
- Line 113. Please add link/reference to the document where information about Mythen 1K linear detector can be found.
- Line 114. Please, use just “step” instead “step length”.
- Line 114. “with" is formatted differently from other text, please correct. Please, check formatting throughout the text.
- Line 115. Please, use any identification word like program/software/package/code with first occurrence “PDFgui”.
- Table 1. What is exactly meaning of the parameters a, b, c, α, β, γ? Could you use kind of picture to visualize theme?
- Line 239. “After HNS is refined, the average length of covalent 239 bonds increase by 0.05 Å”. What was the parameters used for HNS crystal structure model? Manuscript would benefit from this additional information as well as from kind of scheme of bound changes in HNS.
Provided that the above suggestions are taken into account, the paper can be recommended for publication.
Round 2
Reviewer 1 Report
The revised manuscript shows significant improvements however it still requires revision.
1. The title ‘Effects of Particle Size and Thermal Aging on Atomic Scale Local Structures of Ultra-fine Explosives by X-ray Total Scattering’ does not represent the content since particles were modified by re-crystallization and the observed effects may originates namely from the crystallization routine, but not only from particle size.
2. The revised Section 2.2 describes the experimental setup for Total scattering measurements, but the second paragraph (lines 108-115) is doubtful:
‘Diffractometer was used for PDF experiment and the Q resolution is generally very high which should be lower than 0.1% given relatively low energy and extremely high angle resolution in Mythen detector (approximately 0.004°). The Qbroad and Qdamp were set to be zero in fitting r range lower than 10 Å . Standard sample for Qboad calibration was not used, because the limitation in the Qmax that is lower than 18 Å -1 due to the relatively low X-ray energy. Usually for high crystalline material, the too low cutoff Qmax leads to bigger broadening of peak and this make the Qbroad parameter unprecise in calibration.’
Indeed, the Mythen gives perfect angular resolution in reciprocal space, but the overall pattern resolution depends on both detector and goniometer resolution and precision, since one need to move the detector to cover the entire angular range. Anyway, the angular resolution in reciprocal space defines Qdamp parameter of resulted PDF curve. As for the Qbroad, this parameter reflects the broadening of peaks on generated PDF curve. Both parameters are used to be calibrated against external Standard Reference Material. The Qbroad is an important parameter for simulation of theoretical PDF which is required for fitting (Fig. 4B and 4d), 7b and 7d). In general calibration for low Qmax is possible, see for instance the neutron PDF with Qmax ~ 13 (J. Appl. Cryst. (2018). 51, 1492–1497, https://doi.org/10.1107/S1600576718010002) or laboratory Mo-PDF with Qmax ~ 17 (J. Appl. Cryst. (2022). 55, 890-900, https://doi.org/10.1107/S1600576722005878).
3. Lines 116-118: ‘When software GudrunX [37] was used to process the experimental data to obtain the structure factor and PDF curve of all samples, the parameters were optimized for each sample separately’
Please clarify this statement. Generally, PDF significantly depends on parameters which are used for generation from total scattering data. It is not correct to discuss the fine differences between PDFs for different samples in case the parameters were tuned separately for each sample.
4. Lines 119-122: ‘For thermal aging, the heating temperature was 120 ℃, and the aging time was 30 days. Recent experimental studies have found that the particles of ultra-fine explosives grow up after heating at 120 ℃ for 30 days, and the specific surface area decreases significantly. In order to explore its growth mechanism, relevant thermal aging experiments were carried out.’
Please give the references.
Reviewer 2 Report
The manuscript can be published now.
Author Response
The authors would like to thank the reviewers for comments.
Reviewer 3 Report
I accept the manuscript in the present form.
Author Response

(The authors gave the same response as above.)
